# Features of Changes in the Surface Structure and Phase Composition of the of α + β Titanium Alloy after Electromechanical and Thermal Treatment

Vyacheslav Petrovich Bagmutov [1], Valentin Ivanovich Vodopyanov [1], Igor Nikolaevich Zakharov [1], Alexander Yurievich Ivannikov [2,*], Artem Igorevich Bogdanov [1], Mikhail Dmitrievich Romanenko [1] and Vladislav Valerievich Barinov [1]

1   Resistance of Materials Department, Volgograd State Technical University, 400005 Volgograd, Russia
2   Baikov Institute of Metallurgy and Material Science, Russian Academy of Sciences, 119334 Moscow, Russia
*   Correspondence: ivannikov-a@mail.ru

**Abstract:** Changes of structure, phase composition and microhardness in high-strength α + β titanium alloy were investigated after electromechanical treatment (EMT) and subsequent aging. The EMT was performed with alternating current density ($j$ of 300 and 600 A/mm$^2$). The aging was performed upon heating at 600 °C and exposure for 14 h. Methods of scanning electron microscopy, X-ray phase structure, micro X-ray spectral and durometric analyses were used. Modified layers (up to 200–250 μm in depth) on the surface of α + β titanium alloy were formed due to intensive thermo-deformation during the EMT and the following aging. The structure of the surface layer was characterized by high dispersivity (with particle sizes of about 30–500 nm), significant concentration heterogeneity (due to redistribution of the alloying elements and changes in the volume ratio of α- and β-phases), distortions in the crystal structure, increased levels of microstrain and microstress (2–2.5 times as compared to the initial) and increases in microhardness (by 30–40%).

**Keywords:** electromechanical treatment; EMT; alpha-beta; titanium alloy; heat treatment; aging; microstructure; phase analysis; microhardness





## 1. Introduction

Titanium alloys, having high strength, crack resistance, corrosion resistance and resistance to fatigue, are among the most effective structural materials for aviation, shipbuilding, machine building and other industries [1–3]. At the same time, these alloys are characterized by low antifriction properties, a tendency toward scuffing and seizure and low contact endurance. Increasing the above-mentioned performance characteristics is possible through various methods of hardening of the surfaces of titanium alloys, such as thermal [4], chemical–thermal methods, coating [5–8], laser cladding [9–11], laser shock peening [12], plasma spraying [13] and ultrasonic treatment [14–16]. Combined techniques of surface hardening, which join the advantages of various technologies and allow the material hardening capabilities to be realized that cannot be achieved by each of the methods separately, are becoming increasingly widespread. In particular, combinations of intensive heat treatment, laser hardening [17,18], ultrasonic effects [19], shot blasting [20], etc., are used to increase the strength and durability of various titanium alloys. The authors in [21,22] established that the combination of oxidation with ultrasonic hardening increased the depth and hardness of the modified layer. The authors in [23–25] showed the capabilities of combined electromechanical and ultrasonic treatment to increase the static and cyclic strength of pseudo-α and α + β titanium alloys by creating a hardened layer on the surface and by forming compressive residual stress.

In papers [23,24], the structure and mechanical behavior of titanium pseudo-α alloy were explored after various combined treatments. It was shown that the hardening of

alloys with reduced content of the β-phase during high-speed electro–thermal–mechanical influences occurred mainly in the deformation mechanism.

In α + β titanium alloys (also known as VT22 in the Russian Federation), there are more opportunities to regulate the ratio of α- and β-phases, their alloying degree and particle dispersion obtained by various combinations of heat and deformation treatment modes of the alloy. Due to this, the potential for controlling the structure and properties of such alloys during the combined treatment is fully realized in this work. For example, the paper [25] shows the possibility of increasing the strength and fatigue properties of the examined α + β-alloy during surface electromechanical treatment.

The study of the characteristics of high-temperature alloys based on titanium, aluminum, vanadium, molybdenum and others, as well as their mechanical properties and structure, is given considerable attention in recent studies [26–30].

The purpose of the work is to study the specific formation of the microstructure, phase and chemical composition, as well as the microhardness level of the metal surface after combined treatment, which includes intensive thermo-deformation influence during electromechanical hardening and subsequent aging of titanium alloys of the transition class (based on α + β titanium alloy).

## 2. Experimental Procedure

Samples for the tests were made from a α + β titanium alloy bar with a diameter of 20 mm after hot rolling and heat treatment according to the following scheme:

-	heating at 820 °C, exposure τ for 1 h, furnace cooling to 750 °C, exposure τ for 3 h, air cooling;
-	heating at 600 °C, exposure τ for 4 h, air cooling.

The alloy bar was taken as supplied, and the chemical composition of the as-received α + β titanium alloy is given in Table 1.

**Table 1.** Chemical composition of the α + β titanium alloy (wt.%, as per specifications).

| Ti | Al | Mo | V | Cr | Fe | $O_2$ | Others |
|---|---|---|---|---|---|---|---|
| 82.68 | 5.32 | 4.88 | 4.92 | 0.98 | 1 | 0.127 | 0.093 |

The initial properties (according to the specifications) of the alloy are as follows:

-	ultimate strength $\sigma_c$—1150 MPa;
-	yield strength $\sigma_{0.2}$—1060 MPa;
-	residual elongation at break δ—21%;
-	residual reduction of area at break ψ—63%.

Electromechanical treatment (EMT) was used to modify the surface layer. EMT consisted (Figure 1) of exposing alternating current of high density $j$ (300 A/mm$^2$ and 600 A/mm$^2$) and of low voltage $U$ (1.5–2 V) to the material. The current passed through the local contact zone of the tool with the surface of the sample [23–25]. The tool and the treated surface were cooled with water.

The EMT parameters that allowed us to illustrate the most characteristic effects of treatment of α + β titanium alloy were used for the investigated samples:

-	current density $j$—300 and 600 A/mm$^2$;
-	processing speed $V$—0.6 m/min;
-	feed rate $S$—0.2 mm/rev;
-	deforming force $F$—150 N;
-	electrode-tool—tapered rollers from tungsten carbide WC-6Co with diameter of 60 mm, sharpening angle of 5° and contact strip of 0.5–0.7 mm.

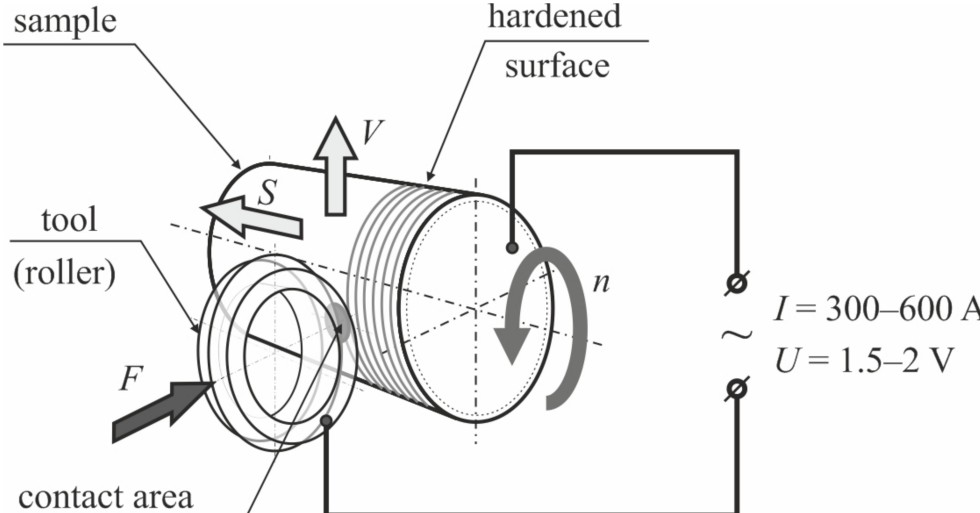

**Figure 1.** Schematic diagram of the electromechanical treatment processing; *V*: processing speed; *S*: feed rate; *F*: deforming force; *I*: current; *U*: voltage; *n*: rotation speed; the arrows indicate direction of rotation, velocity, feed and force vectors.

Aging of the samples after EMT was carried out at heating to 600 °C and exposure $\tau$ for 14 h in the furnace in atmospheric air. The combination with the preceding thermo-deformation treatment gives sufficient increase in strength while maintaining the viscosity properties [31–33].

In the present work, samples of five series were investigated:

- series 1—initial state (as received);
- series 2—EMT with current density 300 A/mm$^2$;
- series 3—EMT with current density 600 A/mm$^2$;
- series 4—series 2 + aging at 600 °C for 14 h;
- series 5—series 3 + aging at 600 °C for 14 h.

On the cylindrical sample surface, a flat area was made with sandpaper to a depth of 0.3–0.5 mm. In order to exclude the effects of post-grinding accretions, the surface of the flat area was etched with 2.5%HNO$_3$–2.5%HF–95%H$_2$O compound.

Measurements of microhardness of the samples surface were carried out on a PMT-3M (LOMO PLC, Saint Petersburg, Russia) microhardness tester with a load of 0.5 N on an indenter and an exposure time of 10 s. A series of 10 marks was applied, and each diagonal of the mark was measured not less than five times until the error of the measurements was less than 1%. The hardware–software complex included a microhardness tester, an optical microscope, a computer and VideoTest-Structure software, allowing for statistical processing of the results.

Metallographic studies of the micro- and macrostructure were carried out on a Metam LV-32 (LOMO PLC, Saint Petersburg, Russia) optical microscope and a FEI Versa 3D Dual-Beam (Thermo Fisher Scientific Inc., Hillsboro, OR, USA) scanning electron microscope.

The chemical composition of separated sections was established on the basis of energy dispersive X-ray spectral analysis using an INCA X-Max spectrometer (Oxford Instruments PLC, Abingdon, Oxfordshire, UK) based on Versa3D.

X-ray structural analysis was performed on a Bruker D8 Advance (Bruker AXS GmbH, Karlsruhe, Germany) X-ray diffractometer. The lattice period of the β-phase was determined on the $\beta_{Ti}$ (110) line, and that of the α-phase was determined on the $\alpha_{Ti}$ (002) and $\alpha_{Ti}$ (101) lines. Semi-quantitative phase analysis was studied using the corundum number method [34], which allows one to quantify the phase ratio of the titanium alloy with an accuracy of about 2–3%.

### 3. Experimental Results

#### 3.1. Macro- and Microstructure

The structures of the as-received samples (Figure 2) were typical for transition-type titanium alloy after significant deformation in the β-phase—the shape of the β-grains was elongated along the rolling direction. The α-phase lamellae edge was thinned (plate thickness was less than 1 μm, and the maximum length was about 10–20 μm; Figure 2b) and was practically invisible on macrosection (Figure 2a); this occurred at high cooling rates. The β-grain size along the rolling direction reached values of more than 500 μm [25].

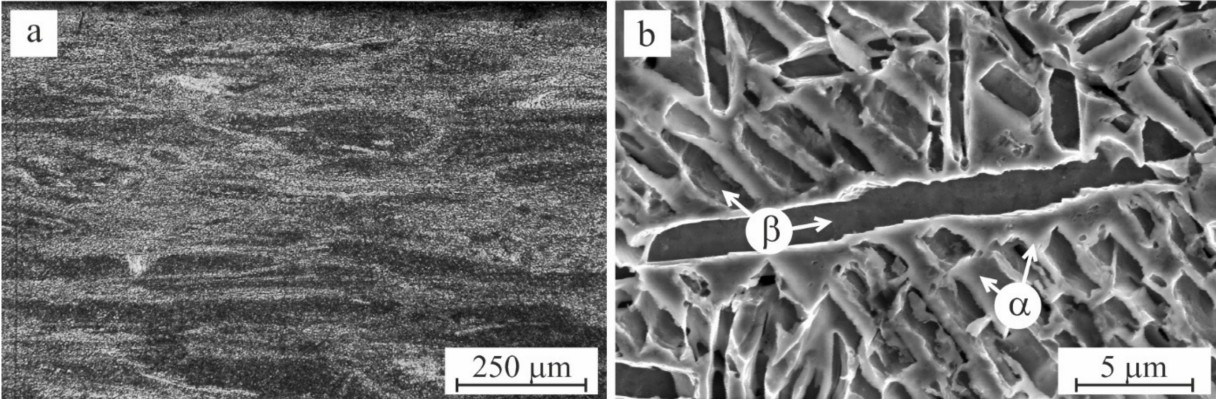

**Figure 2.** Metal structure of as-received samples of α + β titanium alloy (series 1); (**a**) surface layer; (**b**) α- and β-grains.

The features of the microstructure changes along the depth of the sample surface layer after EMT are shown in Figure 3. Three characteristic zones were established: zone 1 was a thin oxidized layer resulting from surface oxidation of the cylindrical sample during EMT (Figure 3, zone 1); zone 2 corresponded to a metal area transformed during thermal and strain effects during EMT (Figure 3, zone 2); zone 3 reflected the initial structure of the titanium alloy (Figure 3, zone 3).

The initial metal (Figures 2 and 3, zone 3) was characterized by a complex pattern. The features of the pattern depended on the orientation of individual fragments of the structure, and this structure had different morphologies in the neighboring microareas. For example, the α-phase plates, which were parallel to the rolling direction, were stretched in this direction, and those perpendicular to it were deformed and curved, acquiring in the process a complex configuration (Figures 2b and 3, zone 3) [25].

The samples of series 2 (after EMT with $j$ = 300 A/mm$^2$) demonstrated a decrease of the primary β-grains extent, indistinct boundaries and disorientation relative to the initial state in the hardened zone (Figure 3a; zone 2). In some areas, the morphology of the particles may have approached a globular morphology, with sizes of 80–400 nm. In general, the observed microstructure was characterized by the basket weaving type. The depth of this zone for the indicated EMT mode was 80–100 μm.

The same characteristic areas (1—surface layer; 2—hardened layer; 3—original metal) were observed in the surface layer of samples of the series 3 (after EMT with $j$ = 600 A/mm$^2$) (Figure 3b). The depth of the hardened zone 2 in this case could reach 200–250 μm. The structure of the hardened by EMT surface layer had a similar pattern in the form of crushed and deformed grains with traces of the original texture both for $j$ = 300 A/mm$^2$ and for $j$ = 600 A/mm$^2$, but in the latter case, the degree of crushing and distortion was noticeably higher (Figure 3a,b).

After aging the microstructure of the surface layer (that previously has been hardened by the EMT) revealed microdispersed particles (Figure 3c,d), which had structured ranging from needle-like (mainly for EMT with current $j$ = 300 A/mm$^2$; Figure 3c) to globular ($j$ = 600 A/mm$^2$; Figure 3d). The boundary between the hardened surface layer and the base metal became almost undetectable.

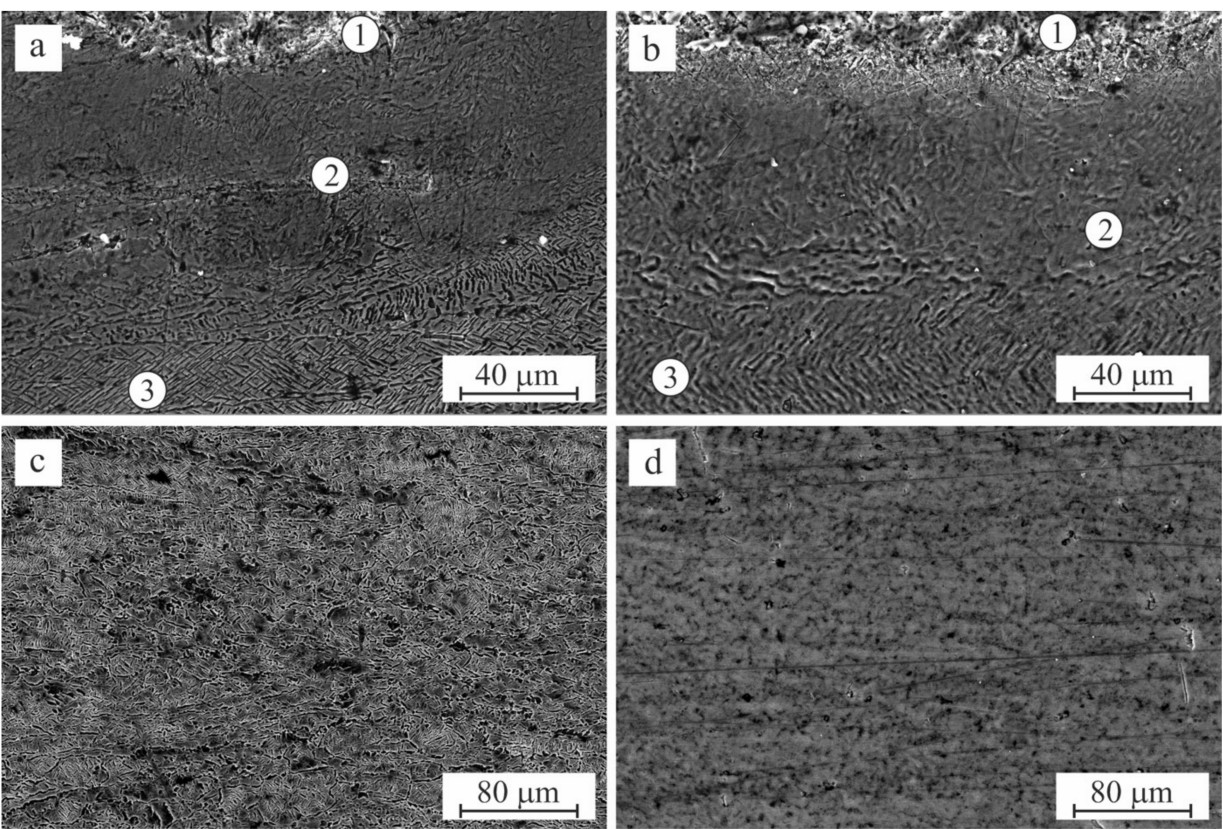

**Figure 3.** Microstructure of the surface layer of the α + β titanium alloy after EMT (**a**) $j$ = 300 A/mm$^2$; (**b**) $j$ = 600 A/mm$^2$; (**c**) $j$ = 300 A/mm$^2$ + aging; (**d**) $j$ = 600 A/mm$^2$ + aging; 1—sample surface (oxide film); 2—near-surface layer after EMT (series 2); 3—initial structure of the titanium alloy (as received).

The results of electron microscopy of these areas at high resolution are shown in Figure 4.

Structures consisting of crushed and deformed primary α- and β-grains were observed in the hardened layer after the EMT with current $j$ = 300 A/mm$^2$. The boundaries between them were visibly blurred, while individual ultradispersed particles of the second phases with dimensions of 50–500 nm were weakly distinguishable within the primary grains (Figure 4a).

Subsequent aging after EMT ($j$ = 300 A/mm$^2$) in the hardened surface layer contributed to the evolution in the structure of β-grains of finely dispersed differently oriented needle-shaped α-phase particles up to 1.5 μm long and up to 100 nm wide, and globules 100–500 nm in diameter with clear boundaries between microstructure elements (Figure 4b).

Figure 4c,d present fragments of the alloy microstructure after EMT with current density $j$ = 600 A/mm$^2$. As can be seen, this treatment mode led to the formation of a recrystallized structure. The boundaries of primary α- and β-grains were almost completely blurred, and small globular particles were also weakly visible (Figure 4c).

β-grains of close to equiaxial polyhedral shapes with sizes up to 30 μm were observed in some areas of the surface layer (Figure 4d). Some grains exhibited intra-grain textures in the form of packets of thin α-plates, which is characteristic of deformation in the (α + β)-area during hardening (Figure 4d). In some grains there were areas with α-phase "zigzag" structures or residual lamellar martensite with ferm-shaped arrangements of plates, which indicated partial hardening from the β-region [31].

The aging of samples at 600 °C after EMT with current $j$ = 600 A/mm$^2$ promoted disintegration of the metastable solid β-solution and release of dispersed α-particles of different morphologies, for example, irregular shapes of 100–500 nm (Figure 4e) or (which prevailed) needle or plate shapes up to 1–1.5 μm long and 30–100 nm thick (Figure 4e,f). At

the same time, the boundaries of some large β-grains (as in Figure 4d) could be preserved as "rims" of α-particles.

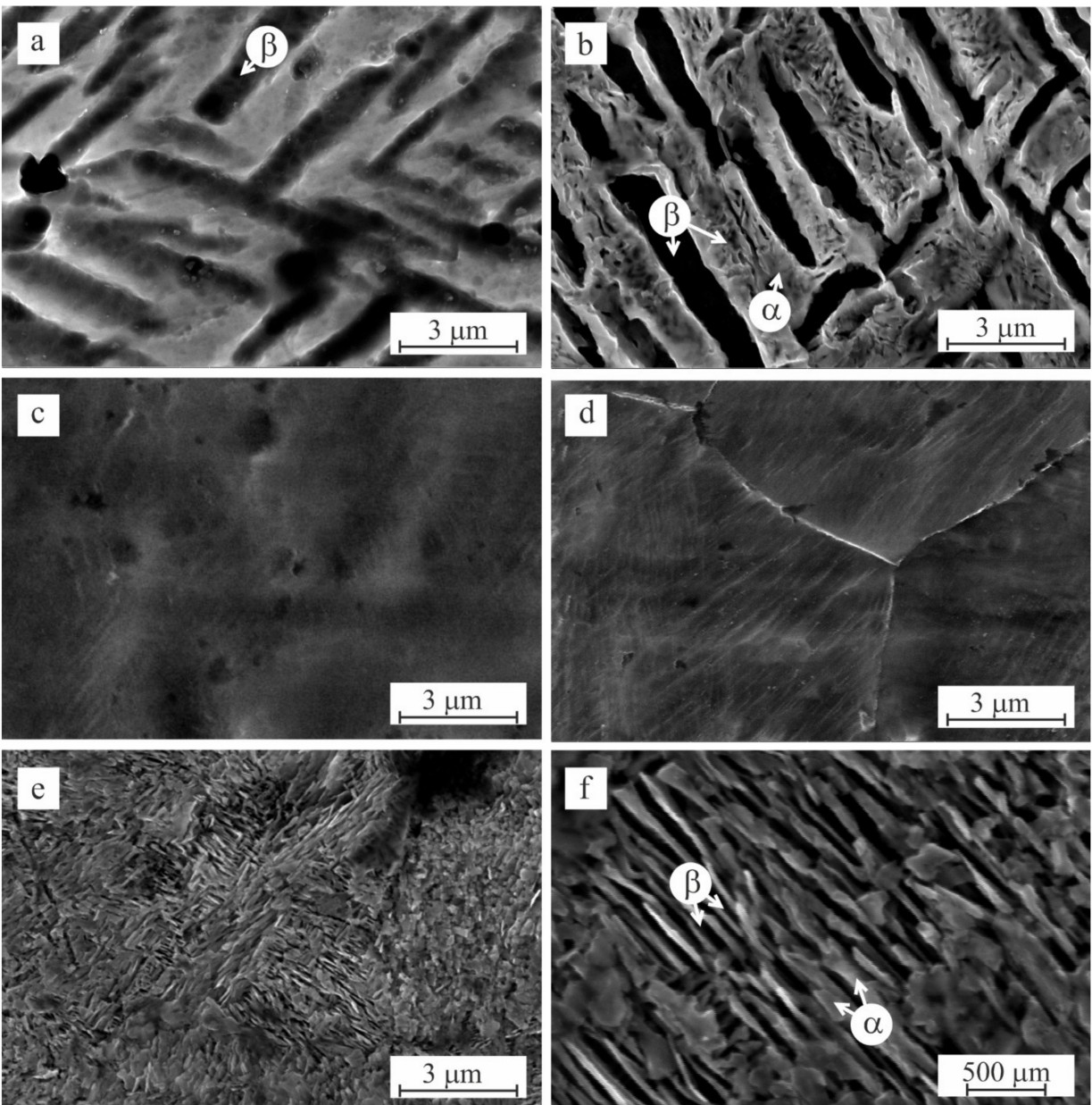

**Figure 4.** Microstructure of the α + β titanium alloy after EMT: (**a**)—after EMT with $j$ = 300 A/mm$^2$; (**b**)—after EMT with $j$ = 300 A/mm$^2$ + aging at 600 °C for 14 h; (**c,d**)—after EMT with $j$ = 600 A/mm$^2$; (**e,f**)—after EMT with $j$ = 600 A/mm$^2$ + aging at 600 °C for 14 h.

### 3.2. Phase Analysis

The results of determining the phase composition of the samples are shown in Table 2.

It could be observed that in the initial state (as received), the volume fraction of the α-phase was 68%, and that of the β-phase was 32%. In the surface layer after EMT hardening, both for $j$ = 300 A/mm$^2$ and $j$ = 600 A/mm$^2$, the volume fractions of phases were approximately the same: 53–55% for α-phase and 45–47% for β-phase. The subsequent aging of the samples after EMT was accompanied by changes of the phase ratio—the α-phase was 68%, while the β-phase was 32%. The phase ratio after EMT and aging was actually the same as for the as-received samples.

**Table 2.** Results of semi-quantitative phase analysis.

| No. of the Treatment Mode Series | Quantity (%) | |
| --- | --- | --- |
| | $\alpha$-Phase | $\beta$-Phase |
| 1—initial state | 68 | 32 |
| 2—EMT with $j$ = 300 A/mm$^2$ | 53 | 47 |
| 3—EMT with $j$ = 600 A/mm$^2$ | 55.4 | 44.6 |
| 4—EMT with $j$ = 300 A/mm$^2$ + aging at 600 °C | 67.8 | 32.2 |
| 5—EMT with $j$ = 600 A/mm$^2$ + aging at 600 °C | 68.4 | 31.6 |

Figure 5 shows diffractograms of the $\alpha$ + $\beta$ titanium alloy after treatment with different modes. It can be seen that the intensity and width of diffraction lines changed as a result of the thermo-deformation influence of EMT. Thus, the intensity of $\beta_{Ti}$ (110) decreased, and the intensity of $\alpha_{Ti}$ (102) and $\alpha_{Ti}$ (101) increased (Figure 5b,c) with respect to the initial state (Figure 5a). The diffractograms show increases in the relative intensity of diffraction lines of the $\beta$-phase after EMT hardening. In series 2 and 3, the $\alpha_{Ti}$ (112) and $\alpha_{Ti}$ (201) lines almost completely disappeared after EMT and appeared again after aging (series 5,6). Decreases in the intensity of $\alpha_{Ti}$ (110) reflexes could be noted after EMT.

With subsequent aging, as could be seen, on the contrary, the intensity of $\beta$-phase lines decreased, and the intensity of $\alpha$-phase lines increased (Figure 5d,e), though the diffractograms of these samples were identical to those of the original samples. With subsequent aging, the return of intensities of these reflexes was noted.

The primary and secondary $\alpha$-phases of titanium have similar types of crystal lattices with their characteristic parameters [32]. In this context, a full-field analysis was not performed in this work because of the difficulty of separating the superimposed diffraction lines of these phases on the diffractogram.

As shown in the diffractograms (Figure 5b,c), the shift of $\beta$-phase lines after EMT occurred in the direction of smaller 2θ angles. This shift corresponded to an increase of the lattice parameter a$_\beta$. The lattice parameter a$_\beta$ is determined by the concentration of the $\beta$-stabilizer substitution elements [32,33]. An increase of the a$_\beta$ parameter characterizes a decrease of the concentration of $\beta$-stabilizers.

The diffractograms after the EMT and subsequent aging of the hardened samples (Figure 5d,e) showed a shift of the $\beta$-phase reflexes towards larger 2θ angles. Thus, subsequent aging was accompanied by a decrease of the a$_\beta$ parameter, i.e., by an increase of the alloying elements' concentrations of the $\beta$-phase and decreases of their volume fractions [33].

Figure 6 shows the results obtained through the analysis of the chemical composition in individual areas of $\alpha$- and $\beta$-grains of the hardened surface based on energy dispersive X-ray spectral analysis (using the INCA X-Max spectrometer based on Versa3D).

The intensity of redistribution of $\beta$-stabilizers between the phases was estimated by the value of the $\beta$-stabilization coefficient $k_\beta$ [31,34]. For the initial state, $k_\beta$ of the $\beta$-phase $k_\beta^\beta$ was 1.69, and the coefficient of the $\alpha$-phase $k_\beta^\alpha$ was 0.37. After the EMT, when the volume fraction of the $\beta$-phase increased from 32% to 47% (Table 2), the $k_\beta^\beta$ was 1.16–1.28 and $k_\beta^\alpha$ was 0.52–0.7.

Analyzing the phases composition, we can conclude that the total amount of the alloying elements in the $\alpha$-phase was 1.5–1.6 times less than in the $\beta$-phase for the initial material. Furthermore, their distribution among the elements was heterogeneous and multidirectional (Figure 6).

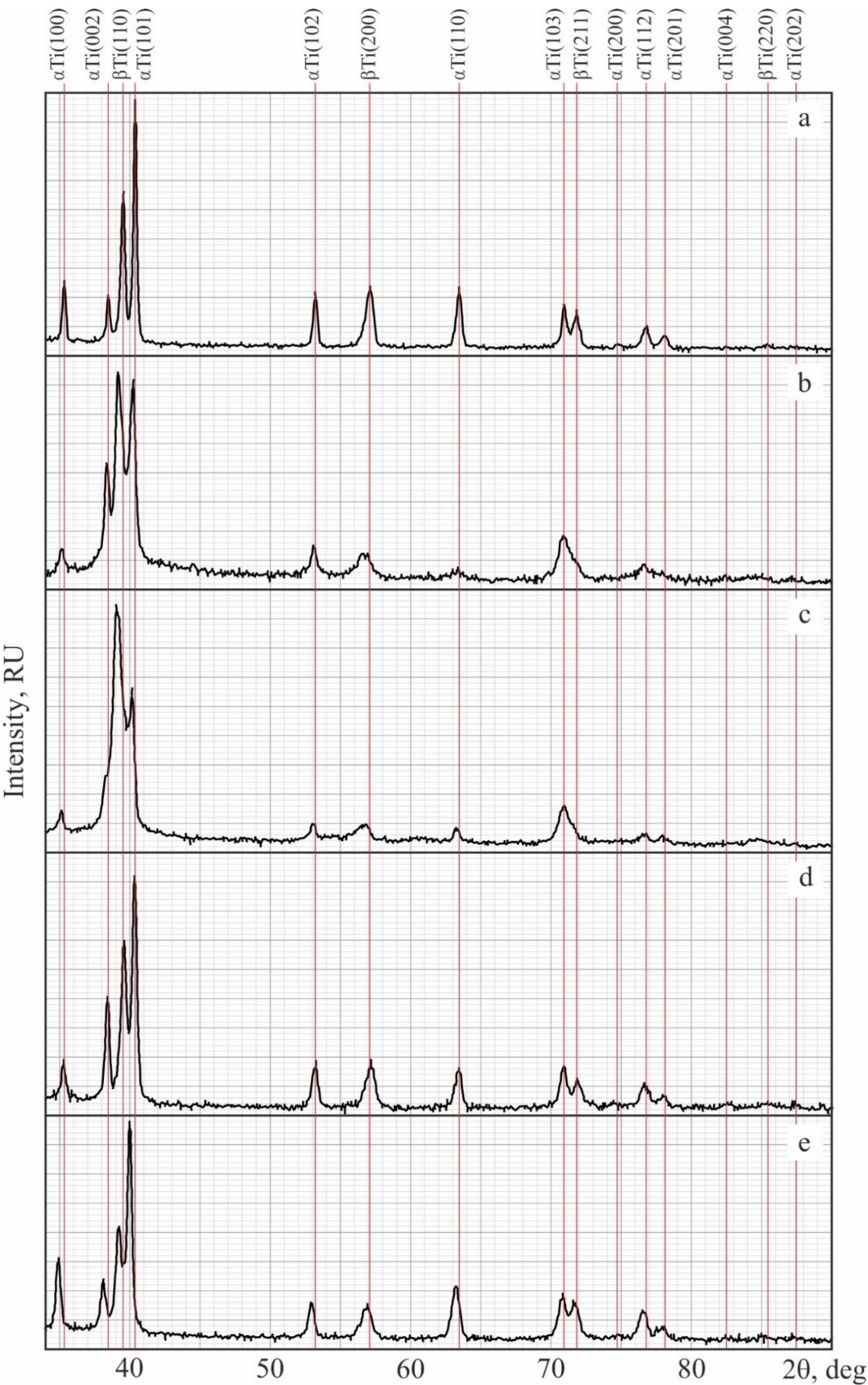

**Figure 5.** Diffractograms of the α + β titanium alloy after various treatments (Table 2): (**a**)—series 1; (**b**)—series 2; (**c**)—series 3; (**d**)—series 4; (**e**)—series 5.

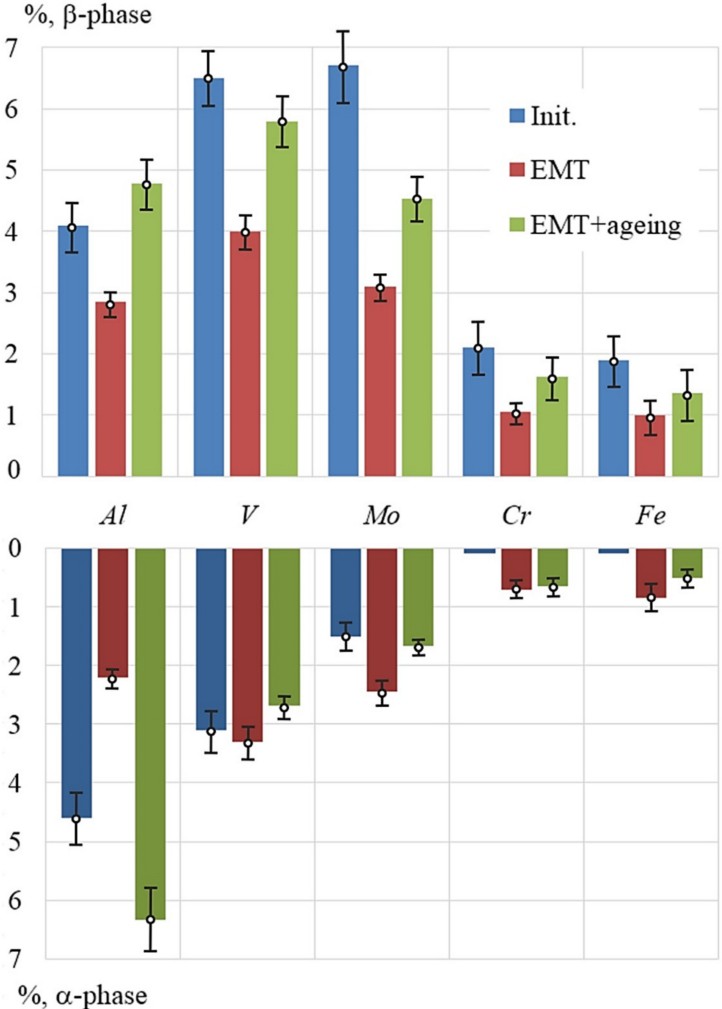

**Figure 6.** Micro-X-ray spectral analysis of the chemical composition of the surface layer after various treatment combinations.

The EMT resulted in redistribution of chemical elements of the α- and β-phases. The increase of the β-phase volume, which was observed during EMT, led to decreases of the β-stabilizers' concentrations. The content of β-stabilizing elements (*V*, *Mo*, *Cr*, *Fe*) in the β-phase decreased by 1.5–2 times in comparison with the initial state (Figure 6).

The chemical compositions of the β- and α-phases practically did not change depending on the current density changes during the EMT.

Aging after EMT led to redistribution of chemical elements in the phases (Figure 6). In the β-phase, the content of all elements increased: *Al*—from 2.85% to 4.75%, *V*—from 4 to 5.785%, *Mo*—from 3.1 to 4.54%, etc., and in the α-phase, only the *Al* content increased, namely, from 2.2% to 6.33% (Figure 6). Comparing the data in Table 2 and Figure 6 indicated that higher contents of the β-phase in the hardened layer structure corresponded to a lower volume fraction of β-stabilizing elements. The above results obtained by two different methods confirmed the nature of redistribution of the alloying elements during EMT.

The chemical analysis also showed that during hardening by EMT and by EMT with subsequent aging, the surface layer of the metal was saturated with oxygen. In the near-surface layer with a thickness of about 10 μm, the oxygen content could reach 5–6% after EMT and 8–13% after EMT and aging.

From the experimental diffractograms, the integral width of the reflexes was calculated by approximation, and the characteristics of the fine crystal structure were determined using the known dependences [34]. The calculated results are given in Table 3.

**Table 3.** The sizes of coherent scattering regions and values of microdistortions of the α- and β-phases in the investigated samples.

| No. of Series | α–Phase | | | β–Phase | | |
|---|---|---|---|---|---|---|
| | $D$ (Å) | $\Delta d/d$ | $\sigma_{II}$ (MPa) | $D$ (Å) | $\Delta d/d$ | $\sigma_{II}$ (MPa) |
| Control | 1975 | $5 \times 10^{-5}$ | 5 | 2874 | $5 \times 10^{-5}$ | 5 |
| 1 | 1004 | 0.00058 | 64 | 2912 | 0.00021 | 24 |
| 2 | 416 | 0.0009 | 98 | 330 | 0.00039 | 43 |
| 3 | 244 | 0.00158 | 177 | 200 | 0.00044 | 49 |
| 4 | 435 | 0.00066 | 73 | 460 | 0.00044 | 50 |
| 5 | 376 | 0.00153 | 171 | 302 | 0.00047 | 52.5 |

It follows from the analysis of the obtained data that the maximum values of microstrains and microstresses were typical for the α-phase of the investigated titanium alloy. Furthermore, the highest levels of microstrains (~0.0016) and microstresses (~180 MPa) corresponded to the EMT with $j$ = 600 A/mm$^2$. After the EMT with $j$ = 300 A/mm$^2$, these values were almost two times lower (0.0009 and 98 MPa, respectively). The aging performed after EMT led to a decrease of these parameters by 25–30% for treatments with current $j$ = 300 A/mm$^2$ and by 3–4% for treatments with $j$ = 600 A/mm$^2$. For the β-phase, regardless of the type of treatment, rather low values of microdistortions (~0.00045) and microstresses (~40–50 MPa) were preserved (Table 3). This effect was likely due to the presence of a larger number of possible slip systems in the BCC lattice of titanium as compared to the HCP lattice [34].

The dimensions of coherent scattering areas after EMT were reduced significantly in comparison with the initial dimensions: for α-phase, 2–4 times (from 1000 Å to 250–400 Å), and for β-phase, 8–14 times (from 3000 Å to 200–350 Å). Thus, the greater degree of refinement of mosaic blocks corresponded to the higher current density ($j$ = 600 A/mm$^2$) during EMT. With subsequent aging, the specified parameters increased 1.3–1.5 times, remaining essentially lower than in the initial level: 2.5–3 times for α-phase and 6–10 times for β-phase (Table 3). Such decreases may indicate the formation of highly defective structures of the surface layer after EMT, in analogy with steels [35].

*3.3. Microhardness*

Table 4 shows average microhardness, standard deviation and relative change compared to the initial state of the samples surface layer (from 20 to 300 μm in depth) of the α + β titanium alloy as a result of hardening by EMT and by EMT with subsequent aging.

**Table 4.** Comparison of microhardness at different treatment modes.

| Sample Series | Treatment Mode | Microhardness (MPa) | Deviation (MPa) | Relative Change |
|---|---|---|---|---|
| 1 | Initial (as received) | 4150 | 318 | 1 |
| 2 | EMT 300 | 4000 | 210 | 0.96 |
| 3 | EMT 600 | 3408 | 329 | 0.82 |
| 4 | EMT 300 + aging | 5436 | 310 | 1.31 |
| 5 | EMT 600 + aging | 5698 | 420 | 1.37 |

As can be seen, after using EMT, including plastic deformation, high-speed heating and cooling of the surface layer, the microhardness decreased by 16–20% compared to the initial state of the sample surface. Such a decrease was characteristic for all EMT modes ($j$ = 300 and $j$ = 600 A/mm$^2$), and only the depth of the layer with the lowered hardness differed. The aging of the samples, which were preliminary hardened by the schemes of the EMT, increased the microhardness of the surface layer by 35–40% (up to 5500–5700 MPa) relative to the initial level (4150 MPa). The aging of samples in the initial state (series 1) did not change the microhardness of the metal.

## 4. Discussion

### 4.1. Effect of EMT on Structure

The results presented in the paper show that the modified layer is formed on the surface of the α + β titanium alloy after EMT. Its structure is characterized by high dispersion, significant concentration heterogeneity and distortions of the crystal structure. According to the results of mathematical modeling [36], there are extremely high temperatures (up to 1200–1500 °C) and heating rates ($10^5$–$10^6$ °C/s) in the local processing area. This contributes to the simultaneous nucleation of a large number of centers of new phases due to heterogeneity, defects and structural imperfections, as well as fluctuations in the concentration of chemical elements in the metal structure. The fixation of the formed ultradispersed particles occurs due to the high cooling rates of the metal ($10^4$–$10^5$ °C/s) [36] and the lack of opportunity for significant growth of these nuclei due to their limitations to the adjacent nuclei, grain boundaries and concentration heterogeneity. Metal melting does not occur in this case due to the very short duration of the exposure of high temperatures (less than 10 ms).

In this case, the higher the current density during EMT is, the higher the temperature in the treatment zone is (if ultra-high rates of heating and cooling remain the same) and, consequently, the higher the flowing intensity of these transformations is. Thus, after the EMT with $j$ = 600 A/mm$^2$, the boundaries of the primary phases are almost completely "blurred" (Figure 4c), and the homogeneity and dispersion of the formed particles increase, both in comparison with the original structure (Figure 2b) and with EMT with current $j$ = 300 A/mm$^2$ (Figure 4a).

### 4.2. Effect of EMT on Microhardness

The described structure changes after EMT are accompanied by some decrease of the microhardness of the treated surface (Table 4). This indicates the incomplete realization of structural strengthening mechanisms and the predominance of unstrengthening processes due to the reduction of the β-phase alloying degree and increase of its volume fraction. The decrease of the β-stabilizing elements' concentrations reduces the stability of the β-phase and leads to its decomposition during cooling and subsequent aging [37]. The layer thickness (100–300 µm) with reduced microhardness corresponds to the size of the areas with altered structures (Figure 3a,b). Thus, in a thin (20–30 microns) near-surface layer, the microhardness can increase 1.5–2 times as compared to the initial state (on some surface areas up to 8000–10,000 MPa) due to the high temperatures, which exceed critical values for an alloy (in some local volumes—up to temperatures of melting, as evidenced by the appearance of areas with recrystallized structures, Figure 4d) and also possible saturation of the surface by oxygen.

### 4.3. Effect of Aging after EMT on Structure and Microhardness

The subsequent aging of samples after EMT increases the microhardness of the surface layer. The average microhardness (in surface depth) increases in comparison, with an initial condition from 12% (after EMT with $j$ = 300 A/mm$^2$ + aging) up to 37% (after EMT with $j$ = 600 A/mm$^2$ + aging). Increases of microhardness in this case can be provided due to joint action of concentration and structural mechanisms of hardening. On the one hand, there is a redistribution of volume fractions of α- and β-phases (Table 2) with increases in concentration in the β-phase of alloying β-stabilizers, which harden the metal. On the other hand, the mixed structure with increased β-phase content obtained in the surface layer during EMT is transformed in the process of aging with the formation of a secondary α-phase in the solid solution of the β-phase in the form of nano- and ultradispersed plate packages of different orientations (Figure 4b,e,f). Such a structure is characterized by sufficiently high strength and hardness [38].

The depth of the layer hardened after aging with increased hardness goes up with increasing current density at the preceding EMT. This is associated with change in the morphology of the forming surface structure, which differs significantly for the EMT modes

$j = 300$ A/mm$^2$ (Figure 4b) and $j = 600$ A/mm$^2$ (Figure 4e). At approximately the same quantitative increase of the $\alpha$-phase volume fraction in both cases, the sizes of the $\alpha$-plates for samples after EMT with current $j = 600$ A/mm$^2$ are smaller and are equally distributed over the surface layer volume, whereas they are formed within the primary $\beta$-grains, preserving the initial heterogeneous structure after EMT with current $j = 300$ A/mm$^2$.

The phase analysis, carried out in combination with the study of the macro- and microstructure of the transition-type $\alpha + \beta$ titanium alloy, allowed us not only to confirm the known laws of hardening of such alloys during heat treatment, but also to detail the features of the EMT and post-EMT aging effect on the ultrafine dispersed structure parameters, the alloying elements' contents in $\alpha$- and $\beta$-phases and microhardness changes of the surface layer and base metal during the combined treatment. It should be noted that it is the realization of the high-gradient system of extreme temperature force fields during the EMT and aging that creates additional opportunities for opening the potential of service properties of the $\alpha + \beta$ titanium alloy.

### 5. Conclusions

The results of microstructure study, micro-X-ray spectral analysis, X-ray phase analysis and microhardness measurements show the possibilities of sequential transformation of structural phase states and strength of the near-surface layer of $\alpha + \beta$ titanium transition class alloy bars in the combined process of EMT with alternating current and of aging.

1. At the first stage of the combined action, the EMT with alternating current leads to the formation of complex multimodal gradient microstructures of titanium alloy characterized by dissolution of the boundaries of primary phases by increased dispersion of formed particles (with a decrease in their size from 2–4 to 8–14 times), by the increased level of microdeformation and microstrain (from 1.5–3 to 6–10 times), by the increased volume fraction of the metastable $\beta$-phase (by 30–40%).

2. The operation carried out after EMT aging is accompanied by the transformation of the mixed surface structure obtained at the previous stage with the formation of nano- and ultradispersed $\alpha$-plates of different orientations in the solid $\beta$-phase solution and saturation of the $\beta$-phase with strengthening elements, which leads to increases of surface microhardness (by 30–35%).

3. The combination of EMT and aging realizes simultaneously the concentration and the structural hardening of the $\alpha + \beta$ titanium alloy surface. The targeted choice of combined treatment modes allows us to regulate the surface strength level according to the microhardness parameter due to the creation of structures with the given distribution of alloying elements (according to the $\beta$-stabilization coefficient), with the given ratio between the volumes of $\alpha$- and $\beta$-phases, and the degree of dispersion of surface particles.

**Author Contributions:** Supervision, validation, funding acquisition V.P.B.; conceptualization, methodology, writing—original draft preparation, V.I.V. and I.N.Z.; surface treatment, investigation of the structure, writing—review and editing, I.N.Z. and A.Y.I.; supervision, A.Y.I.; SEM analysis, M.D.R. and V.V.B.; X-ray analysis, M.D.R. and A.I.B.; preparation of the specimens, V.V.B. All authors have read and agreed to the published version of the manuscript.

**Funding:** This research was funded by Russian Science Foundation (project No. 22-29-01078).

**Institutional Review Board Statement:** Not applicable.

**Informed Consent Statement:** Not applicable.

**Data Availability Statement:** Data sharing not applicable.

**Conflicts of Interest:** The authors declare no conflict of interest.

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
