# Peer review of "Features of Changes in the Surface Structure and Phase Composition of the of α + β Titanium Alloy after Electromechanical and Thermal Treatment"

_metals, doi:10.3390/met12091535_

Round 1

Reviewer 1 Report

In this manuscript, the authors have used electromechanical and thermal treatment to study the surface structure and phase composition of α+β Titanium alloy. The authors have found that the targeted choice of combined treated models allow to regulated the surface strength. These results are very interesting. It is acceptable for publication in this journal after revision.

1.The abstract should be given the specific data to shows the research purpose.

2.As we know, the overall performances of high-temperature alloys are influenced by the mixed phases. I think that we should be focused on the overall performances of a single phase rather than mixed phase. Therefore, why the authors consider the α+β Titanium alloys? If the composition has many phases, how to the overall performances of Ti-Al alloy is influenced by these mixed phases?

3.In introduction, the high-temperature alloys should be well summarized. For the structural stability of high-temperature Al-based alloys, the authors should be cited these references: Philos Mag. 2022;102:1386-1399. Int J Quantum Chem. 2022;122:e26825. J Mater Eng Perform. 2021;30:8289-8295. J Mater Eng Perform. 2021;30:2661-2668. Ceram Int. 2022;48:11518-11526.

4.As the high-temperature alloys, we focus on the mechanical or other properties, have you consider the mechanical or the other properties for Ti-Al alloys?

5.Experimental procedures, we know that the quality of sample is strongly related to the vacuum degree. So what the degree of vacuum? In particular, the authors how to examine the gas(H and O) effect.

6.In Figure 3, the authors should be given the distribution of various phases.

7.What the role of α or β phase in Ti-Al alloys?

Author Response

Dear Reviewer,

Thank you for your useful comments and suggestions on the structure of our manuscript.

  1. Comment: The abstract should be given the specific data to shows the research purpose..

Response: We agree with the comment. The following corrections have been made to the text of the abstract:

Changes of structure, phase composition and microhardness in high-strength α+β titanium alloy were investigated after electromechanical treatment (EMT) and subsequent aging. The EMT was performed with alternating current density (j) of 300 and 600 A/mm2. The aging was performed upon heating at 600 ºC and exposure for 14 hours. Methods of scanning electron microscopy, X-ray phase structure, micro X-ray spectral and durometric analyses were used.

Modified layer (up to 200 – 250 µm in depth) was formed on the surface of α+β titanium alloy due to intensive thermal deformation during the EMT and the following aging. The structure of the surface layer is characterized by high dispersivity (with particle size of about 30 – 500 nm), significant concentration heterogeneity (due to redistribution of the alloying elements and changes in the volume ratio of α- and β-phases), distortions in the crystal structure, increased level of microstrain and microstress (2 – 2.5 times as compared to the initial), as well as an increase in microhardness (by 30-40%).

  1. Comment: As we know, the overall performances of high-temperature alloys are influenced by the mixed phases. I think that we should be focused on the overall performances of a single phase rather than mixed phase. Therefore, why the authors consider the α+β Titanium alloys? If the composition has many phases, how to the overall performances of Ti-Al alloy is influenced by these mixed phases?

Response: We agree with the comment. In the text (lines 42-49) the following explanations are made to the choice of α+β-alloy for research:

In papers [23, 24] the structure and mechanical behavior of titanium pseudo-α alloy were explored after various combined treatments. It was shown that the hardening of alloys with reduced content of β-phase during high-speed electro-thermo-mechanical influences occurs mainly in the deformation mechanism.

In α+β titanium alloys (also known as VT22 in Russian Federation) there are more opportunities to regulate the ratio of α- and β-phases, their alloying degree, the particle dispersion, obtained by various combinations of heat and deformation treatment modes of the alloy. Due to this, the potential of controlling the structure and properties of such alloys during the combined treatment is fully realized in this work. For example, the paper [25] shows the possibility of increasing the strength and fatigue properties of the examined α+β-alloy during surface electromechanical treatment.

  1. Comment: In introduction, the high-temperature alloys should be well summarized. For the structural stability of high-temperature Al-based alloys, the authors should be cited these references: Philos Mag. 2022;102:1386-1399. Int J Quantum Chem. 2022;122:e26825. J Mater Eng Perform. 2021;30:8289-8295. J Mater Eng Perform. 2021;30:2661-2668. Ceram Int. 2022;48:11518-11526..

Response: We agree with the comment. The literature review has been supplemented with references to the mentioned works (insert before line 50):

The study of the characteristics of high-temperature alloys based on titanium, aluminum, vanadium, molybdenum and others, their mechanical properties and structure is given considerable attention in modern studies [26-30].

[26]     PAN Y. First-Principles Investigation of the Structural, Mechanical, and Thermodynamic Properties of Hexagonal and Cubic MoAl5 Alloy [J]. Journal of Materials Engineering and Performance, 2021, 30: 8289–8295.

[27]     PAN Y. Insight into the Mechanical Properties and Fracture Behavior of Pt3Al Coating by Experiment and Theoretical Simulation [J]. Journal of Mate-rials Engineering and Performance, 2021, 30: 2661–2668.

[28]     PAN Y. New insight into the structural, mechanical, electronic, and thermodynamic properties of the monoclinic TMAl3-type aluminides [J]. International Journal of Quantum Chemistry, 2022, 122 (2): e26825.

[29]     PAN Y. Influence of vacancy on the mechanical and thermodynamic properties of PtAl2 from the first-principles investigation [J]. Philosophical Magazine. Part A: Materials Science, 2022, 102 (14): 1386-1399.

[30]     PU D., PAN Y. First-principles investigation of oxidation mechanism of Al-doped Mo5Si3 silicide [J]. Ceramics International, 2022, 48 (8): 11518-11526.

  1. Comment: As the high-temperature alloys, we focus on the mechanical or other properties, have you consider the mechanical or the other properties for Ti-Al alloys?

Response: We agree with the comment. At this stage of the study the estimation of strength properties of the near-surface layer was carried out according to change of the microhardness value. The paper [25] shows possibilities to improve strength and fatigue properties of the examined α+β-alloy during surface electromechanical treatment. In the following publications we’ll investigate tribotechnical properties of this alloy after surface hardening.

  1. Comment: Experimental procedures, we know that the quality of sample is strongly related to the vacuum degree. So what the degree of vacuum? In particular, the authors how to examine the gas (H and O) effect.

Response: We agree with the comment. All technological operations of metal hardening were carried out in atmospheric air. The access of oxygen and nitrogen in the process is technologically limited by the closed contact zone of the tool and the sample, high heating and cooling rates. The chemical analysis (performed by using an INCA X-Max spectrometer based on Versa3D) showed that some saturation of the surface layer with oxygen occurs during the process of metal hardening – the content of oxygen in the near-surface layer (which is about 10 μm thick) can reach from 5 – 6 % to 8 – 13 %.

The experimental studies of the alloy structure on a Versa3D DualBeam scanning electron microscope were carried out using deep (high) vacuum, the studies on a Bruker D8 Advance X-ray diffractometer were carried out in atmospheric air. The preparation of the samples surface for the X-ray analysis (grinding, polishing, etching) was carried out immediately before the experiment. No reflexes from oxide and other phases were detected on the diffractograms.

  1. Comment: In Figure 3, the authors should be given the distribution of various phases.

Response: We agree with the comment. The picture has been improved.

  1. Comment: What the role of α or β phase in Ti-Al alloys?

Response: In this work the role of individual phases in formation of structure and properties was not investigated. It is shown that during the intensive electro-thermal treatment of the surface of titanium α+β alloys the greatest strengthening effect is achieved when forming the metal structure with uniform distribution in the volume of primary grains of α- and β-phase ultradispersed particles.

Reviewer 2 Report

The article is well planned. Experiment well described and reproducible. The selected research methods are adequate to the tested materials. The description of the research results is clear and does not raise any major objections.

Comments:

- are the authors sure of the accuracy of the given phase composition (table 2)?

- how were the samples prepared for XRD tests?

- how many repetitions were made for the microhardness? Are the authors sure of such a high accuracy of the measurements?

- authors should check references to tables and figures. I think line 269 should be table 3.

Author Response

Dear Reviewer,

Thank you for your useful comments and suggestions on the structure of our manuscript.

1. Comment: are the authors sure of the accuracy of the given phase composition (table 2)?

Response: We agree with the comment. An explanation has been added to the description of the experiment methodology (line 110):

Semi-quantitative phase analysis was studied using the corundum number method, which allows to quantify the phase ratio of the titanium alloy with an accuracy of about 2 – 3 %.

  1. Comment: how were the samples prepared for XRD tests?

Response: We agree with the comment. An explanation has been added to the description of the experiment methodology (line 110):

On the cylindrical sample surface, a flat area was made with sandpaper to a depth of 0.3 – 0.5 mm. In order to exclude the effects of post-grinding accretions, the surface of the flat area was etched with 2.5%HNO3–2.5%HF–95%H2O compound.

  1. Comment: how many repetitions were made for the microhardness? Are the authors sure of such a high accuracy of the measurements?

Response: We agree with the comment. An explanation has been added to the description of the experimental methods (lines 94-98):

Measurements of microhardness of the samples surface were carried out on a PMT-3M microhardness tester with a load of 0,5 N on an indenter and exposure time of 10 s. A series of 10 marks was applied, each diagonal of the mark was measured not less than five times until the error of measurements was less than 1 %.

  1. Comment: authors should check references to tables and figures. I think line 269 should be table 3.

Response: We agree with the comment. Necessary corrections have been made to the text.

Round 2

Reviewer 1 Report

The authors have made required changes. I recommend it for publication in its current form.